# TreNet: Hybrid Neural Networks for Learning the Local Trend in Time Series

**Tao Lin,**[*] **Tian Guo**[*] **& Karl Aberer**
School of Computer and Communication Sciences
Ecole polytechnique federale de Lausanne
Lausanne, Switzerland
`{tao.lin, tian.guo, karl.aberer}@epfl.ch`

## Abstract

Local trends of time series characterize the intermediate upward and downward patterns of time series. Learning and forecasting the local trend in time series data play an important role in many real applications, ranging from investing in the stock market, resource allocation in data centers and load schedule in smart grid. Inspired by the recent successes of neural networks, in this paper we propose TreNet, a novel end-to-end hybrid neural network that predicts the local trend of time series based on local and global contextual features. TreNet leverages convolutional neural networks (CNNs) to extract salient features from local raw data of time series. Meanwhile, considering long-range dependencies existing in the sequence of historical local trends, TreNet uses a long-short term memory recurrent neural network (LSTM) to capture such dependency. Furthermore, for predicting the local trend, a feature fusion layer is designed in TreNet to learn joint representation from the features captured by CNN and LSTM. Our proposed TreNet demonstrates its effectiveness by outperforming conventional CNN, LSTM, HMM method and various kernel based baselines on real datasets.

## 1 Introduction

Time series, which is a sequence of data points in time order, is being generated in a wide spectrum of domains, such as daily fluctuation of the stock market, power consumption records of households, performance monitoring data of clusters in data centres, and so on. In many applications, users are interested in understanding the evolving trend in time series and forecasting the trend, since the conventional prediction on specific data points could deliver very little information about the semantics and dynamics of the underlying process generating the time series. For instance, time series in Figure 1 are from the household power consumption dataset[1]. Figure 1(a) shows some raw data points of time series. Though point $A$ and $B$ have approximately the same value, the underlying system is likely to be in two different states when it outputs $A$ and $B$, because $A$ is in an upward trend while $B$ is in a downward trend (Wang et al., 2011; Matsubara et al., 2014). On the other hand, even when two points with the similar value are both in the upward trend, e.g, point $A$ and $C$, the different slopes and durations of the trends where point $A$ and $C$ locate, could also indicate different states of the underlying process.

Particularly, in this paper we are interested in the local trend of time series which measures the intermediate local behaviour, i.e., upward or downward pattern of time series that characterized by the slope and duration (Wang et al., 2011). For instance, in Figure 1(b) the linear segments over raw data points of time series represent the local trends extracted from a real household power consumption time series. For the ease of presentation, we will use the term trend and local trend interchangeably in the rest of the paper. Learning and forecasting local trends are quite useful in a wide range of applications. For instance, in the stock market, due to its high volatility and noisy environment, in reality predicting stock price trends is preferred over the prediction of the stock market absolute values (Atsalakis & Valavanis, 2009). Predicting the local trend of stock price time series empowers

---

[*]These two authors contributed equally.
[1] https://archive.ics.uci.edu/ml/datasets/Individual+household+electric+power+consumption

traders to design profitable trading strategies (Chang et al., 2012b; Atsalakis & Valavanis, 2009). In the smart energy domain, knowing the predictive local trend of power consumption time series enables energy providers to schedule power supply and maximize energy utilization (Zhao & Magoulès, 2012).

Meanwhile, in recent years neural networks have shown the dramatical power in a wide spectrum of domains, e.g., natural language processing, computer vision, speech recognition, time series analysis, etc. (Wang et al., 2016b; Sutskever et al., 2014; Yang et al., 2015; Lipton et al., 2015). For time series data, two mainstream architectures, convolutional neural network (CNN) and recurrent neural network (RNN) have been exploited in different time series related tasks, e.g., RNN in time series classification (Lipton et al., 2015) and CNN in activity recognition and snippet learning (Liu et al., 2015; Yang et al., 2015). RNN is powerful in discovering the dependency in sequence data (Jain et al., 2014; Graves, 2012) and particularly the Long Short-Term Memory (LSTM) RNN works well on sequence data with long-term dependencies (Chung et al., 2014; Hochreiter & Schmidhuber, 1997) due to the internal memory mechanism. CNN excels in exacting effective representation of local salience from raw data of time series by enforcing a local connectivity between neurons. (Yang et al., 2015; Hammerla et al., 2016).

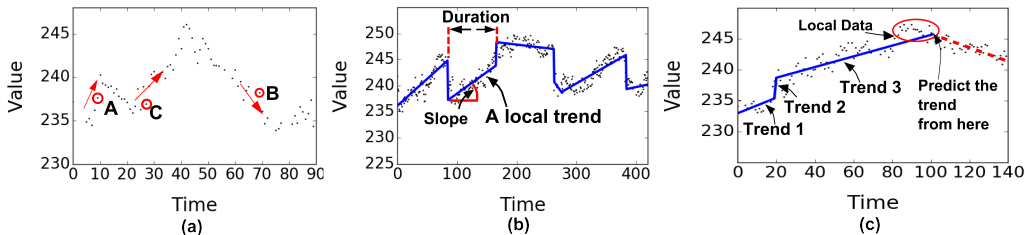

Figure 1: (a) Time series of household power consumption. (b) Local trends in time series. (c) Effect of local raw data on the trend forecasting.

In this paper, we focus on learning and forecasting the local trends in time series via neural networks. This involves learning different aspects of the data. On one hand, the sequence of historical local trends describes the long-term contextual information of time series and thus naturally affects the evolution of the following local trend. On the other hand, the recent raw data points of time series (Wang et al., 2011; Batal et al., 2012), which represent the local variation and behaviour of time series, affect the evolving of the following trend as well and have particular predictive power for abruptly changing local trends (Wang et al., 2011). For instance, in Figure 1(c), trend 1, 2 and 3 present a continuous upward pattern. Then when we aim at predicting the subsequent trend of time series at the end of the third local trend, the previous three successive upward trends outline a probable increasing trend afterwards. However, the local data around the end of the third trend, e.g., data points in the red circle, indicate that time series could stabilize and even decrease. The data points after the third trend indeed present a decreasing trend indicated by the red dotted segment. In this case, the subsequent trend has more dependency on the local data points. Therefore, it is highly desired to develop a systematic way to model such various hidden and complementary dependencies in time series for the local trend forecasting problem.

To this end, we propose a end-to-end hybrid neural network, referred to as TreNet. In particular, it consists of a LSTM recurrent neural network to capture the long dependency in historical local trends, a convolutional neural network to extract local features from local raw data of time series, and a feature fusion layer to learn joint representation to take advantage of both features drawn from CNN and LSTM. Such joint representation is used for the local trend forecasting. The experimental analysis on real datasets demonstrates that TreNet outperforms individual recurrent neural network, convolutional neural network and a variety of baselines in term of local trend prediction accuracy.

The rest of the paper is organized as follows. Section 2 presents related work, while Section 3 defines the problem to be solved and introduces the notations. In Section 4, we present the proposed TreNet. Section 5 demonstrates the performance of our method and baselines on real datasets. Finally, the paper is concluded in Section 6. Refer to Section 7 and Section 8 for more experiment results and discussion.

## 2 RELATED WORK

Traditional learning approaches over local trends of time series mainly make use of Hidden Markov Models (HMMs) (Wang et al., 2011; Matsubara et al., 2014). HMMs maintain short-term state dependences, i.e., the memoryless Markov property and predefined number of states, which requires significant task specific knowledge. RNNs instead use high dimensional, distributed hidden states that could take into account long-term dependencies in sequence data. Previous time series segmentation approaches (Keogh et al., 2001; Matsubara et al., 2014; Yuan, 2015) focus on achieving a meaningful segmentation and finding patterns, rather than modeling the relation in segments and therefore are not suitable for forecasting local trends. Multi-step ahead prediction is another way to realize local trend prediction by fitting the predicted values to estimate the local trend. However, multi-step ahead prediction is a non-trivial problem itself (Chang et al., 2012a). In this paper, we concentrate on directly learning local trends through neural networks.

RNNs have recently shown promising results in a variety of applications, especially when there exist sequential dependencies in data (Lyu & Zhu, 2014; Chung et al., 2014; Sutskever et al., 2014). Long short-term memory (LSTM) (Hochreiter & Schmidhuber, 1997; Lyu & Zhu, 2014; Chung et al., 2014), a class of recurrent neural networks with sophisticated recurrent hidden and gated units, are particularly successful and popular due to its ability to learn hidden long-term sequential dependencies. (Lipton et al., 2015) uses LSTMs to recognize patterns in multivariate time series, especially for multi-label classification of diagnoses. (Chauhan & Vig, 2015; Malhotra et al., 2015) evaluate the ability of LSTMs to detect anomalies in ECG time series. Bidirectional LSTM (Graves & Schmidhuber, 2005) is usually intended for speech processing rather than time series forecasting problems. Our paper focuses on using LSTM to capture the dependency in the sequence of historical local trends and meanwhile the hidden states in LSTM are further used to learn joint feature representations for the local trend forecasting.

CNN is often used to learn effective representation of local salience from raw data (Vinyals et al., 2015; Donahue et al., 2015; Karpathy et al., 2014). (Hammerla et al., 2016; Yang et al., 2015; Lea et al., 2016) make use of CNNs to extract features from raw time series data for activity/action recognition. (Liu et al., 2015) focuses on the prediction of periodical time series values by using CNN and embedding time series with the potential neighbors in the temporal domain. Our proposed TreNet will combine the strengths of both LSTM and CNN and form a novel and unified neural network architecture for local trend forecasting.

Hybrid neural networks, which combines the strengths of various neural networks, are receiving increasing interest in the computer vision domain, such as image captioning (Mao et al., 2014; Vinyals et al., 2015; Donahue et al., 2015), image classification (Wang et al., 2016a), protein structure prediction (Li & Yu, 2016), action recognition (Ballas et al., 2015; Donahue et al., 2015) and so on. But efficient exploitation of such hybrid architectures has not been well studied for time series data, especially the trend forecasting problem. (Li & Yu, 2016; Ballas et al., 2015) utilize CNNs over images in cascade of RNNs in order to capture the temporal features for classification. (Bashivan et al., 2015) transforms EEG data into a sequence of topology-preserving multi-spectral images and then trains a cascaded convolutional-recurrent network over such images for EEG classification. (Wang et al., 2016a; Mao et al., 2014) propose the CNN-RNN framework to learn a shared representation for image captioning and classification problems. In our proposed TreNet, LSTM and CNN first respectively learn the trend evolution and local raw data of time series and then TreNet fuses the features captured by LSTM and CNN to predict the trend.

## 3 PROBLEM FORMULATION

In this section, we provide the formal definition of the trend learning and forecasting problem in this paper.

We define time series as a sequence of data points $\mathcal{X} = \{x_1, \ldots, x_T\}$, where each data point $x_t$ is real-valued and subscript $t$ represents the time instant. The corresponding *local trend sequence* of $\mathcal{X}$ is a series of piecewise linear representations of $\mathcal{X}$, denoted by $\mathcal{T} = \{\langle \ell_k, s_k \rangle\}$. Each element of $\mathcal{T}$, e.g., $\langle \ell_k, s_k \rangle$ describes a linear function over a certain subsequence (or segment) of $\mathcal{X}$ and corresponds to a local trend in $\mathcal{X}$. Such local trends in $\mathcal{T}$ are extracted from $\mathcal{X}$ by time series segmentation and fitting a linear function *w.r.t.* time $t$ over each segment (Keogh et al., 2001; Wang

et al., 2011). $\ell_k$ and $s_k$ respectively represent the duration and slope of trend $k$. $\ell_k$ is measured in terms of the time range covered by trend $k$. Local trends in $\mathcal{T}$ are time ordered and non-overlapping. The durations of all the local trends in $\mathcal{T}$ address $\sum_k \ell_k = T$. In addition, a local trend sequence ending by time $t$ is denoted by $\mathcal{T}(t) = \{\langle \ell_k, s_k \rangle \mid \sum_k \ell_k \le t\}$.

Meanwhile, as we discussed in Section 1, local raw data of time series affects the varying of trend as well and thus we define the *local data w.r.t.* a certain time instant $t$ as a sequence of data points in a window of size $w$, denoted by $\mathcal{L}(t) = \{x_{t-w}, \dots, x_t\}$.

At certain time $t$, trend forecasting is meant to predict the duration and slope of the following trend based on a given sequence of historical trends $\mathcal{T}(t)$ and local data set $\mathcal{L}(t)$. The predicted duration and slope at time $t$ are denoted by $\hat{\ell}_t$ and $\hat{s}_t$. Our proposed TreNet can be trained for predicting either $\hat{\ell}_t$ or $\hat{s}_t$. For simplicity, we use $\hat{y}_t$ to represent the predicted value of TreNet throughout the paper.

Therefore, given the training dataset $\mathcal{D} = \mathcal{X} \cup \mathcal{T}$, we aim to propose a neural network based approach to learn a function $\hat{y}_t = f(\mathcal{T}(t), \mathcal{L}(t))$ for the trend forecasting. In this paper, we focus on univariate time series. The proposed method can be naturally generalized to multivariate time series as well by augmenting the input to the neural network. Refer to Section 8 for more discussion.

## 4 HYBRID NEURAL NETWORKS FOR TREND LEARNING AND FORECASTING

In this section, we first present an overview about the proposed TreNet for the trend forecasting. Then we will detail the components of TreNet.

**Overview.**
The idea of our TreNet is to combine CNN with LSTM to utilize their representation abilities on different aspects of training data $\mathcal{D}$ ($\mathcal{D} = \mathcal{X} \cup \mathcal{T}$) and then to learn a joint feature for the trend prediction. Technically, TreNet is designed to learn a predictive function $\hat{y}_t = f(R(\mathcal{T}(t)), C(\mathcal{L}(t)))$. $R(\mathcal{T}(t))$ is derived by training the LSTM over sequence $\mathcal{T}$ to capture the dependency in the trend evolving, while $C(\mathcal{L}(t))$ corresponds to local features extracted by CNN from $\mathcal{L}(t)$. The long-term and local features captured by LSTM and CNN, i.e., $R(\mathcal{T}(t))$ and $C(\mathcal{L}(t))$ convey complementary information pertaining to the trend varying. Therefore, the feature fusion layer is supposed to take advantages of both features to produce a fused representation for improved performance. Finally, the trend prediction is realized by the function $f(\cdot, \cdot)$, which corresponds to the feature fusion and output layers in Figure 2.

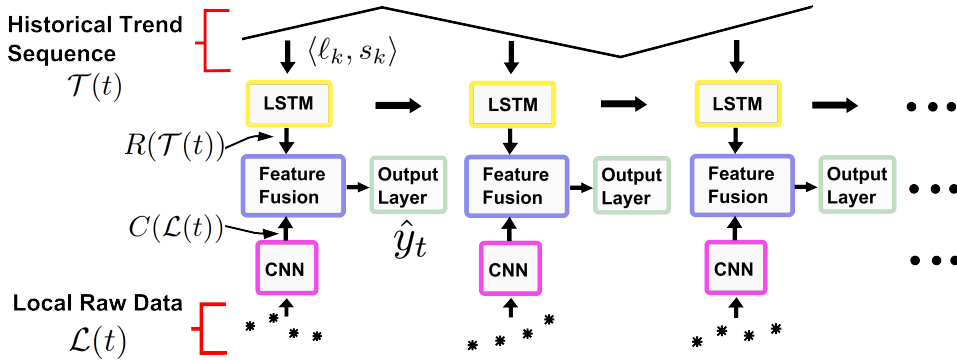

Figure 2: Illustration of the hybrid architecture of TreNet. (best viewed in colour)

**Learning the dependency in the trend sequence.**
During the training phase, the duration $\ell_k$ and slope $s_k$ of each local trend $k$ in sequence $\mathcal{T}$ are fed into the LSTM layer of TreNet. Each $j$-th neuron in the LSTM layer maintains a memory $c_k^j$ at step $k$. The output $h_k^j$ or the activation of this neuron is then expressed as (Hochreiter & Schmidhuber,

1997; Chung et al., 2014):

$$h_k^j = o_k^j tanh(c_k^j) \tag{1}$$

where $o_k^j$ is an output gate and calculated as:

$$o_k^j = \sigma(\boldsymbol{W_o}[\ell_k \ s_k] + \boldsymbol{U_o}\boldsymbol{h_{k-1}} + \boldsymbol{V_o}\boldsymbol{c_k})^j \tag{2}$$

where $[\ell_k \ s_k]$ is the concatenation of the duration and slope of the trend $k$, $\boldsymbol{h_{k-1}}$ and $\boldsymbol{c_k}$ are the vectorization of the activations of $\{h_{k-1}^j\}$ and $\{c_k^j\}$, and $\sigma$ is a logistic sigmoid function. Then, the memory cell $c_k^j$ is updated through partially forgetting the existing memory and adding a new memory content $\tilde{c}_k^j$:

$$c_k^j = f_k^j c_{k-1}^j + i_k^j \tilde{c}_k^j \ , \ \ \tilde{c}_k^j = tanh(\boldsymbol{W_c}[\ell_k \ s_k] + \boldsymbol{U_c}\boldsymbol{h_{k-1}})^j \tag{3}$$

The extent to which the existing memory is forgotten is modulated by a forget gate $f_k^j$, and the degree to which the new memory content is added to the memory cell is modulated by an input gate $i_k^j$. Then, such gates are computed by

$$f_k^j = \sigma(\boldsymbol{W_f}[\ell_k \ s_k] + \boldsymbol{U_f}\boldsymbol{h_{k-1}} + \boldsymbol{V_f}\boldsymbol{c_{k-1}})^j \tag{4}$$

$$i_k^j = \sigma(\boldsymbol{W_i}[\ell_k \ s_k] + \boldsymbol{U_i}\boldsymbol{h_{k-1}} + \boldsymbol{V_i}\boldsymbol{c_{k-1}})^j \tag{5}$$

At each step $k$, the hidden activation $\boldsymbol{h_k}$ is the output to the feature fusion layer. Specifically, given a $\mathcal{T}(t)$ containing $n$ local trends (i.e., $|\mathcal{T}(t)| = n$), the output of $R(\mathcal{T}(t))$ is $R(\mathcal{T}(t)) = \boldsymbol{h_n}$.

**Learning features from the local raw data of time series.**
When the $k$-th trend in $\mathcal{T}$ is fed to LSTM, the corresponding local raw time series data input to the CNN part of TreNet is $\mathcal{L}(t)$, where $t = \sum_{i=1}^{k} \ell_i$. CNN consists of $H$ stacked layers of 1-d convolutional, activation and pooling operations. Denote by $\boldsymbol{a}^i$ the input signal of layer $i$ and thus at the first layer $a^1 = \mathcal{L}(t)$. Each layer has a specified number of filters $n^i$ of a specified filter size $d^i$. Each filter on a layer sweeps through the entire input signal to exact local features as follows:

$$v_m^{i,j} = \phi(b^{i,j} + \sum_{z=m-d^i/2}^{m+d^i/2} W_z^{i,j} a_z^i) , \forall m = 1, \dots, |a^i| \tag{6}$$

where $v_m^{i,j}$ is the activation of $j$-th filter of layer $i$ on $m$ position of the input signal. Here $\phi$ is the Leaky Rectified Linear Unit, which is shown to perform better (Xu et al., 2015). Then the max-pooling is performed over the $v_m^{i,j}$ of each filter.

Finally, the output of CNN in TreNet is the concatenation of max-pooling of each filter on the last layer $H$, namely:

$$C(\mathcal{L}(t)) = [p^1, \dots, p^{n^H}], \ p^j = [\max_{1 \le z \le q}(\{v_{m+z}^{H,j}\})], \ \ \forall j = 1, \dots, n^H \tag{7}$$

where $q$ is the pooling size.

**Feature fusion and output layers.**
The feature fusion layer combines the representations $R(\mathcal{T}(t))$ and $C(\mathcal{L}(t))$, to form a joint feature. Then, such joint feature is fed to the output layer to provide the trend prediction. Particularly, we first map $R(\mathcal{T}(t))$ and $C(\mathcal{L}(t))$ to the same feature space and add them together to obtain the activation of the feature fusion layer (Mao et al., 2014). The output layer is a fully-connect layer following the feature fusion layer. Mathematically, the prediction of TreNet is expressed as:

$$\hat{y}_t = f(R(\mathcal{T}(t)), \ C(\mathcal{L}(t))) = \boldsymbol{W^o} \cdot \underbrace{\phi(\boldsymbol{W^r} \cdot R(\mathcal{T}(t)) + \boldsymbol{W^c} \cdot C(\mathcal{L}(t)))}_{feature \ fusion} + \boldsymbol{b^o} \tag{8}$$

where $\phi(\cdot)$ is element-wise leaky ReLU activation function and $+$ denotes the element-wise addition. $\boldsymbol{W^o}$ and $\boldsymbol{b^o}$ are the weights and bias of the output layer.

To train TreNet, we adopt the squared error function plus a regularization term as:

$$J(\boldsymbol{W}, \boldsymbol{b} \, ; \mathcal{T}, \mathcal{X}) = \frac{1}{|\mathcal{T}|} \sum_{k=1}^{|\mathcal{T}|} (\hat{y}_k - y_k)^2 + \lambda \|\boldsymbol{W}\|_2 \tag{9}$$

where $\boldsymbol{W}$ and $\boldsymbol{b}$ represent the weight and bias parameters in TreNet, $\lambda$ is a hyperparameter for the regularization term and $y_k$ is the true value of trend slope or duration.

The cost function is differentiable and the architecture of TreNet allows the gradients from the loss function (9) to be backpropagated to both LSTM and CNN parts. TreNet can be trained respectively for the slope and duration of local trends using $\mathcal{T}$ and $\mathcal{X}$. When performing forecasting, $\mathcal{T}(t)$ and $\mathcal{L}(t)$ are fed to TreNet and the prediction value $\hat{y}_k$ could be either the slope or duration depending on the training target.

# 5 EXPERIMENTAL ANALYSIS

In this section, we conduct extensive experiments to demonstrate the prediction performance of TreNet by comparing to a variety of baselines. Due to the page limit, refer to Section 7 for more experiment results.

## 5.1 EXPERIMENT SETUP

**Dataset:** We test our method and baselines on three real time series datasets.

- **Daily Household Power Consumption** (HousePC). This dataset[2] contains measurements of electric power consumption in one household with a one-minute sampling rate over a period of almost 4 years. Different electrical quantities and some sub-metering values are available. We use the voltage time series throughout the experiments.

- **Gas Sensor** (GasSensor). This dataset[3] contains the recordings of chemical sensors exposed to dynamic gas mixtures at varying concentrations. The measurement was constructed by the continuous acquisition of the sensor array signals for a duration of about 12 hours without interruption. We mainly use the gas mixture time series regarding Ethylene and Methane in air.

- **Stock Transaction** (Stock): This dataset is extracted from Yahoo Finance and contains the daily stock transaction information in New York Stock Exchange from 1950-10 to 2016-4.

All datasets are preprocessed by (Keogh et al., 2001) to extract local trends. Alternative time series segmentation and local trend extraction approaches can be used as well. We choose (Keogh et al., 2001) here due to its high efficiency. Totally, we obtain 42591, 4720 and 1316 local trends respectively from above datasets. For the ease of experimental result interpretation, the slope of extracted local trends is represented by the angle of the corresponding linear function and thus in a bounded value range $[-90, 90]$. The duration of local trends is measured by the number of data points within the local trend. Then, the obtained trend sequences and the set of local data are split into training (80%), validation (10%) and test (10%) datasets.

**Baselines:** We compare TreNet with the following six baselines:

- **CNN**. This baseline method predicts the trend by only using CNN over the set of local raw data of time series to learn features for the forecasting. The size of local data is set at $w$ as is defined in Section 3.

- **LSTM**. This method uses LSTM to learn dependencies in the trend sequence $\mathcal{T}$ and predicts the trend only using the trained LSTM.

- **Support Vector Regression (SVR)**. A family of support vector regression based approaches with different kernel methods is used for the trend forecasting. We consider three

---

[2] https://archive.ics.uci.edu/ml/datasets/Individual+household+electric+power+consumption
[3] https://archive.ics.uci.edu/ml/datasets/Gas+sensor+array+under+dynamic+gas+mixtures

| Dataset | Model | RMSE @ Duration | RMSE @ Slope |
|---|---|---|---|
| | CNN | 27.51 | 13.56 |
| | LSTM | 27.27 | 13.27 |
| | SVRBF | 31.81 | 12.94 |
| HousePC | SVPOLY | 31.81 | 12.93 |
| | SVSIG | 31.80 | 12.93 |
| | pHMM | 34.06 | 26.00 |
| | Naive | 39.68 | 21.17 |
| | CLSTM | 25.97 | 13.77 |
| | TreNet | 25.89 | 12.89 |
| | CNN | 18.87 | 12.78 |
| | LSTM | 11.07 | 8.40 |
| | SVRBF | 11.38 | 7.40 |
| Stock | SVPOLY | 11.40 | 7.42 |
| | SVSIG | 11.49 | 7.41 |
| | pHMM | 36.37 | 8.70 |
| | Naive | 11.36 | 8.58 |
| | CLSTM | 9.26 | 7.31 |
| | TreNet | 8.86 | 6.84 |
| | CNN | 53.99 | 11.51 |
| | LSTM | 55.77 | 11.22 |
| | SVRBF | 62.81 | 10.21 |
| GasSensor | SVPOLY | 70.91 | 10.95 |
| | SVSIG | 85.69 | 11.92 |
| | pHMM | 111.62 | 13.07 |
| | Naive | 53.76 | 10.57 |
| | CLSTM | 54.20 | 14.86 |
| | TreNet | 52.28 | 9.57 |

Table 1: RMSE of the prediction of local trend duration and slope on each dataset.

commonly used kernels (Liu et al., 2015), i.e., Radial Basis kernel (**SVRBF**), Polynomial kernel (**SVPOLY**), Sigmoid kernel (**SVSIG**). The trend sequence and the corresponding set of local time series data are concatenated as the input features to such SVR approaches.

- **Pattern-based Hidden Markov Model (pHMM)**. (Wang et al., 2011) proposed a pattern-based hidden Markov model (HMM), which segments the time series and models the dependency in segments via HMM. The derived HMM model is used to predict the state of time series and then to estimate the trend based on the state.

- **Naive**. This is the naive approach which takes the duration and slope of the last trend as the prediction for the next one.

- **ConvNet+LSTM(CLSTM)**. It is based on the cascade structure of ConvNet and LSTM in (Bashivan et al., 2015) which feeds the features learnt by ConvNet over time series to a LSTM and obtains the prediction from the LSTM.

**Evaluation metric:** We evaluate the predictive performance of TreNet and baselines in terms of Root Mean Square Error (RMSE). The lower the RMSE, the more accurate the predictions.

**Training:** The training procedure of TreNet and baselines in our paper follows the schema below.

The CNN and LSTM components in TreNet share the same network structure (e.g., number of layers, neurons in each layer) as CNN and LSTM baselines. CNN has two stacked convolutional layers, which have 32 filters of size 2 and 4. The number of memory cells in LSTM is 600. For baseline CNN and LSTM, we tune the learning rate for each approach from $\{10^{-1}, 10^{-2}, 10^{-3}, 10^{-4}, 10^{-5}\}$ (Sutskever et al., 2013), in order to achieve the least prediction errors and then fix the learning rate. For TreNet, in addition to the learning rate, the number of neurons in the feature fusion layer is chosen from the range $\{300, 600, 900, 1200\}$ to achieve the best performance. We use dropout and L2 regularization to control the capacity of neural networks to prevent overfitting, and set the values to $0.5$ and $5 \times 10^{-4}$ respectively for all datasets (Mao et al., 2014). The Adam optimizer (Kingma & Ba, 2014) is chosen to learn the weights in neural networks.

Regarding the SVR based approaches, we carefully tune the parameters $c$ (error penalty), $d$ (degree of kernel function), and $\gamma$ (kernel coefficient) for kernels. Each parameter is selected from the sets $c \in \{10^{-5}, 10^{-4}, \ldots, 1, \ldots, 10^4, 10^5\}$, $d \in \{1, 2, 3\}$, $\gamma \in \{10^{-5}, 10^{-4}, \ldots, 1, \ldots, 10^5\}$ respectively. We iterate through candidate values of each combination of $c$, $d$ and $\gamma$ to train our model, and keep the parameters that generate the lowest RMSE on the validation set, and then use them to predict on the test set.

The training datasets of SVR and pHMM baselines are consistent as that of TreNet. Likewise, CNN and LSTM baselines are respectively fed by the set of local data and the trend sequence of the same size as TreNet. In addition, since the window size of local data is tunable, we vary the window size of local data, i.e. $w$, from the range $\{100, 300, 500, 700, 900\}$, so as to investigate how the size of local data influences the predication performance. The results will be presented in Section 5.2. The model's performance on the validation set will be evaluated after each epoch of training. Each model is trained for at least 50 epochs. Meanwhile, the training process adopts early stopping if no further improvement in the performance of validation shows up after 50 epochs.

## 5.2 EXPERIMENT RESULTS

Table 1 studies the prediction performances of TreNet and baselines. For each dataset, the window size of local data is constant for approaches (i.e., CNN, SVRBF, SVPOLY, SVSIG, pHMM and TreNet) that take local data as input. Then, the results of each approach are obtained by tuning the corresponding parameter as described in Section 5.1.

In Table 1, we observe that TreNet consistently outperforms baselines on the duration and slope prediction by achieving around $30\%$ less errors at the maximum. It verifies that the hybrid architecture of TreNet can improve the performance by utilizing the information captured by both CNN and LSTM. Specifically, pHMM method performs worse due to the limited representation capability of HMM. On the slope prediction, SVR based approaches can get comparable results as TreNet.

In the following group of experiments, we investigate the effect of local data size (i.e., $w$) on the prediction. In particular, we tune the value of local data size for the approaches whose input fea-

tures contains local data and observe the prediction errors. Such approaches include CNN, SVRBF, SVPOLY, SVSIG, pHMM and TreNet. LSTM only consumes the trend sequence and thus is not included. Due to the page limit, we report the results on the HousePC dataset in Table 2 and Table 3. The results on Stock and GasSensor datasets can be referred to Section 7.

Baseline Naive has no original time series data as input CLSTM works on the whole time series and has no local data. Thus they are excluded from this set of experiments.

In Table 2, we observe that compared to baselines TreNet has the lowest errors on the duration prediction across different window sizes. pHMM requires sufficient data points to model the relations of segments and fails to work on 100 size. As the window size increases and more local data points are fed to the training process, the prediction errors of CNN and TreNet decrease or nearly stabilize. This could be because only the certain amount of local data has predictive power. The filtering and pooling mechanism enables CNN to focus on the certain local data having strong predictive power and thus giving more local data only gives rise to marginal improvements. Such similar phenomenon is observed on the slope prediction as is shown in Table 3. For more results and discussion, please refer to Section 7.

| Window Size | CNN | SVRBF | SVPOLY | SVSIG | pHMM | TreNet |
|---|---|---|---|---|---|---|
| 100 | 29.37 | 31.48 | 31.96 | 31.88 | - | 25.93 |
| 300 | 27.33 | 31.17 | 31.61 | 31.66 | 30.03 | 25.94 |
| 500 | 27.51 | 31.81 | 31.81 | 31.80 | 34.06 | 25.89 |
| 700 | 27.41 | 31.10 | 31.09 | 31.11 | 27.37 | 25.72 |
| 900 | 27.42 | 31.28 | 31.27 | 31.27 | 28.45 | 25.62 |

Table 2: RMSE of the duration predictions *w.r.t.* different sizes of local data in HousePC dataset

| Window Size | CNN | SVRBF | SVPOLY | SVSIG | pHMM | TreNet |
|---|---|---|---|---|---|---|
| 100 | 13.68 | 12.93 | 12.9352 | 12.9346 | - | 13.14 |
| 300 | 13.60 | 12.93 | 12.9346 | 12.9345 | 27.75 | 13.15 |
| 500 | 13.56 | 12.94 | 12.9342 | 12.9346 | 26.00 | 12.89 |
| 700 | 13.52 | 12.93 | 12.9345 | 12.9345 | 35.32 | 12.86 |
| 900 | 13.60 | 12.94 | 12.9350 | 12.9346 | 37.60 | 12.96 |

Table 3: RMSE of the slope predictions *w.r.t.* different sizes of local data in HousePC dataset

## 6 CONCLUSION

In this paper we propose TreNet, a novel hybrid neural network to learn and predict the local trend behaviour of time series. The experimental results demonstrate that such a hybrid framework can indeed utilize complementary information extracted by CNN and LSTM to enhance the prediction performance. Moreover, such architecture is generic and extendible in that additional exogenous time series can be fed to TreNet, so as to boost the performance and investigate the effect of different data sources on the trend evolving.

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

## 7  APPENDIX

### 7.1  DATA PRE-PROCESSING

In this part, we describe the data pre-processing, which extracts the local trend sequence from raw time series data for the subsequent neural network training and testing.

We convert the raw time series data into a piecewise linear representation, namely consecutive segments (Keogh et al., 2001; Wang et al., 2011). Each segment corresponds to a local trend and is fitted by a linear function of time series value *w.r.t.* time, e.g., $x_t = \beta_1 t + \beta_0 + \epsilon$ over the time range $[t_1, t_2)$ of this segment. Then, the slope and duration are derived from the coefficient $\beta_1$ and $[t_1, t_2]$.

Technically, we adopt the bottom-up approach in (Keogh et al., 2001), since it can achieve lower approximate errors compared with top-down and sliding window methods. The process is illustrated in Figure 3. Initially, we approximate time series $\mathcal{X}$ with $\lfloor \frac{T}{2} \rfloor$ line segments ($T$ is the length of the

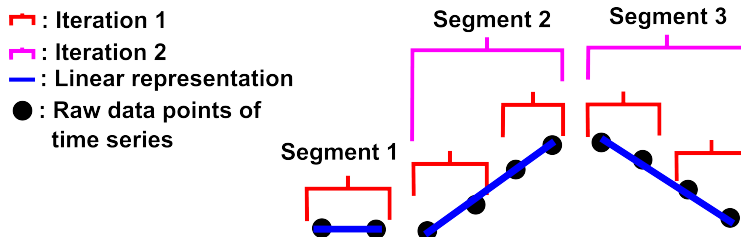

Figure 3: Illustration of local trend extraction via time series segmentation. (Best viewed in colour)

time series). Then, we iteratively merge the neighbouring segments to build longer ones. In each iteration, neighbouring segments with the minimal approximation error are merged into a new one. The merging process repeats until every possible merge gives rise to a segment with errors above a specified threshold. We use the relative mean squared error as the error metric and specify the threshold as $0.05$.

## 7.2 ADDITIONAL EXPERIMENT RESULTS

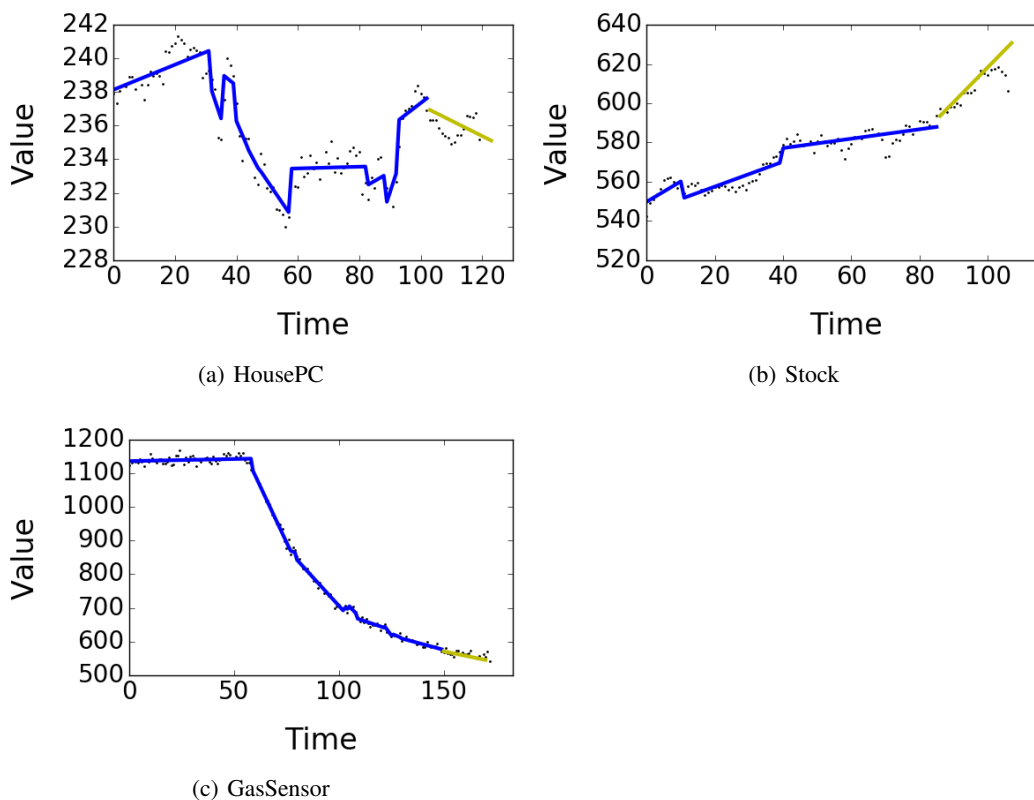

Figure 4: Visualization of the trend prediction by TreNet in HousePC, Stock and GasSensor datasets. The blue line in each figure represents the historical trend sequence. The yellow line represents the predicted local trend.

In this group of experiments, we visualize the trend prediction using the sample testing data instance from each dataset in Figure 4. We can observe that in HousePC TreNet successfully predicts the changed trend, though there are successive upward trends before. In Stock and GasSensor datasets, the succeeding upward and downward trends are correctly predicted as well.

| Window Size | CNN | SVRBF | SVPOLY | SVSIG | pHMM | TreNet |
|---|---|---|---|---|---|---|
| 100 | 18.87 | 11.38 | 11.40 | 11.49 | - | 8.86 |
| 300 | 18.17 | 11.41 | 11.44 | 11.42 | 39.84 | 8.85 |
| 500 | 18.06 | 11.39 | 11.44 | 11.36 | 32.10 | 8.51 |
| 700 | 18.10 | 11.45 | 11.59 | 11.58 | 36.37 | 8.58 |
| 900 | 18.07 | 11.32 | 11.47 | 11.59 | 38.36 | 8.78 |

Table 4: RMSE of the duration predictions on different sizes of local data in Stock dataset

| Window Size | CNN | SVRBF | SVPOLY | SVSIG | pHMM | TreNet |
|---|---|---|---|---|---|---|
| 100 | 12.78 | 7.40 | 7.42 | 7.41 | - | 6.84 |
| 300 | 12.24 | 7.42 | 7.51 | 7.38 | 6.67 | 6.53 |
| 500 | 12.13 | 7.47 | 7.41 | 7.42 | 7.59 | 6.58 |
| 700 | 12.24 | 7.53 | 7.58 | 7.51 | 9.74 | 6.75 |
| 900 | 12.25 | 7.61 | 7.45 | 7.59 | 14.00 | 6.73 |

Table 5: RMSE of the slope predictions on different sizes of local data in Stock dataset

Then, we provide the RMSE *w.r.t.* the varying window size on Stock and GasSensor datasets in Table 4, Table 5, Table 6 and Table 7.

From the results, we observe that TreNet outperforms baselines almost on all window sizes. Meanwhile, the prediction errors often present the decreasing and stable pattern as the window size varies.

**Window size of local data:** The observation in above experiments *w.r.t.* the varying window size provides inspiration for choosing the window size of local data. Given the training dataset, we can find out the maximum duration of local trends and takes it as the local data size. This is because doing so can ensure that the range of local data in each training instance can cover the most recent local trend, whose raw data is believed to have strong predictive power for the subsequent trend. Additionally, we observe that setting the window size of local data of CNN and TreNet in this way can achieve comparable prediction errors compared to the cases with larger window sizes .

| Window Size | CNN | SVRBF | SVPOLY | SVSIG | pHMM | TreNet |
|---|---|---|---|---|---|---|
| 100 | 54.23 | 57.77 | 65.99 | 99.78 | - | 53.91 |
| 300 | 53.99 | 62.81 | 70.91 | 85.69 | - | 52.28 |
| 500 | 53.82 | 61.86 | 64.33 | 91.51 | 111.62 | 51.77 |
| 700 | 53.14 | 61.20 | 63.89 | 78.20 | 175.36 | 51.15 |
| 900 | 53.19 | 61.45 | 63.83 | 68.09 | 255.73 | 51.25 |

Table 6: RMSE of the duration predictions on different sizes of local data in GasSensor dataset

| Window Size | CNN | SVRBF | SVPOLY | SVSIG | pHMM | TreNet |
|---|---|---|---|---|---|---|
| 100 | 11.98 | 11.16 | 11.19 | 12.48 | - | 10.30 |
| 300 | 11.51 | 10.21 | 10.95 | 11.92 | - | 9.57 |
| 500 | 11.75 | 10.08 | 10.65 | 11.64 | 13.07 | 9.60 |
| 700 | 11.59 | 9.54 | 10.44 | 11.72 | 12.29 | 9.55 |
| 900 | 12.10 | 9.61 | 10.37 | 11.54 | 12.37 | 9.46 |

Table 7: RMSE of the slope predictions on different sizes of local data in GasSensor dataset

# 8 DISCUSSION

For multivariate time series, we can augment the input of TreNet by including the trend sequences and local data of exogenous time series and then train TreNet for a certain target time series to predict its trend. Another line of research is to explore equipping TreNet with multi-task learning. This is motivated by the observation that if we decompose the trend forecasting problem into classification and regression respectively for the slope and duration, we can utilize the correlation between slope

and duration to boost the prediction performance. In addition, there could be alternative frameworks to combine the outputs of CNN and LSTM and our work opens the door for applying hybrid neural networks for trend analysis in time series.

