# Peer review of "TreNet: Hybrid Neural Networks for Learning the Local Trend in Time Series"

_ICLR 2017 — rejected_

[Public Comment · (anonymous) · 07 Nov 2016]
**Section 3: About slope and duration calculation**

Interesting work! Thank you for your contributions. It would be great if you explain a bit more on local trend extraction process in Section 3, diagrammatically if possible!

[Author Response · Tao Lin · 01 Dec 2016]
**Updated Version**

Dear Reviewers,

We upload a new version of the paper and list the content updated as follows:

1. Refine Figure 1 and Figure 2.
2. In the experiment section, update some experiment results and the result discussion.
3. In the appendix section, add data pre-processing subsection to explain the local trend extraction from raw time series data. Add the result discussion.

Thanks!

[Official Review · AnonReviewer3 · rating 5 · confidence 5 · 15 Dec 2016 (modified: 20 Jan 2017)]
**Interesting idea of trend prediction but incomplete baselines and experiments.**

Revision of the review:
The authors did a commendable job of including additional references and baseline experiments.

---

This paper presents a hybrid architecture for time series prediction, focusing on the slope and duration of linear trends. The architecture consists of combining a 1D convnet for local time series and an LSTM for time series of trend descriptors. The convnet and LSTM features are combined into an MLP for predicting either the slope or the duration of the next trend in a 1D time series. The method is evaluated on 3 small datasets.

Summary:
This paper, while relative well written and presenting an interesting approach, has several methodology flaws, that should be handled by new experiments.

Pros:
The idea of extracting upward or downward trends from time series - although these should, ideally be learned, not rely on an ad-hoc technique, given that this is a submission for ICLR.

Methodology:
* In section 3, what do you mean by predicting “either [the duration] $\hat l_t$ or [slope] $\hat s_t$” of the trend? Predictions are valid only if those two predictions are done jointly. The two losses should be combined during training.
* In the entire paper, the trend slope and duration need to be predicted jointly. Predicting a time series without specifying the horizon of the prediction is meaningless. If the duration of the trends is short, the time series could go up or down alternatively; if the duration of the trend is long, the slope might be close to zero. Predictions at specific horizons are needed.
* In general, time series prediction for such applications as trading and load forecasting is pointless if no decision is made. A trading strategy would be radically different for short-term and noisy oscillations or from long-term, stable upward or downward trend. An actual evaluation in terms of trading profit/loss should be added for each of the baselines, including the naïve baselines.
* As mentioned earlier in the pre-review questions, an important baseline is missing: feeding the local time series to the convnet and connecting the convnet directly to the LSTM, without ad-hoc trend extraction.
* The convnet -> LSTM architecture would need an analysis of the convnet filters and trend prediction representation.
* Trend prediction/segmentation by the convnet could be an extra supervised loss.
* The detailed analysis of the trend extraction technique is missing.
* In section 5, the SVM baselines have local trend and local time series vectors concatenated. Why isn’t the same approach used for LSTM baselines (as a multivariate input) and why the convnet operates only on local 
* An important, “naïve” baseline is missing: next local trend slope and duration = previous local trend slope and duration.

Missing references:
The related work section is very partial and omits important work in hybrid convnet + LSTM architectures, in particular:
Vinyals, Oriol, Toshev, Alexander, Bengio, Samy, and Erhan, Dumitru. Show and tell: A neural image caption generator. CoRR, abs/1411.4555, 2014.
Donahue, Jeff, Hendricks, Lisa Anne, Guadarrama, Sergio, Rohrbach, Marcus, Venugopalan, Subhashini, Saenko, Kate, and Darrell, Trevor. Long-term recurrent convolutional networks for visual recognition and description. CoRR, abs/1411.4389, 2014.
Karpathy, Andrej, Toderici, George, Shetty, Sanketh, Leung, Thomas, Sukthankar, Rahul, and Fei-Fei, Li. Large-scale video classification with convolutional neural networks. In CVPR, 2014.

The organization of the paper needs improvement:
* Section 3 does not explain the selection of the maximal tolerable variance in each trend segment. The appendix should be moved to the core part of the paper.
* Section 4 is unnecessarily long and gives well known details and equations about convnets and LSTMs. The only variation from standard algorithm descriptions is that $l_k$ $s_k$ are concatenated. The feature fusion layer can be expressed by a simple MLP on the concatenation of R(T(l)) and C(L(t)). Details could be moved to the appendix.

Additional questions:
*In section 5, how many datapoints are there in each dataset? Listing only the number of local trends is uninformative.

Typos:
* p. 5, top “duration and slop”

[Official Review · AnonReviewer2 · rating 6 · confidence 4 · 16 Dec 2016 (modified: 21 Jan 2017)]
**Intriguing problems and architecture but proposed approach not fully justified**

Updated review: the authors did an admirable job of responding to and incorporating reviewer feedback. In particular, they put a lot of effort into additional experiments, even incorporating a new and much stronger baseline (the ConvNet -> LSTM baseline requested by multiple reviewers). I still have two lingering concerns previously stated -- that each model's architecture (# hidden units, etc.) should be tuned independently and that a pure time series forecasting baselines (without the trend preprocessing) should be tried. I'm going to bump up my score from a clear rejection to a borderline.

-----

This paper is concerned with time series prediction problems for which the prediction targets include the slope and duration of upcoming local trends. This setting is of great interest in several real world problem settings (e.g., financial markets) where decisions (e.g., buy or sell) are often driven by local changes and trends. The primary challenge in these problems is distinguishing true changes and trends (i.e., a downturn in share price) from noise. The authors tackle this with an interesting hybrid architecture (TreNet) with four parts: (1) preprocessing to extract trends, (2) an LSTM that accepts those trends as inputs to ostensibly capture long term dependencies, (3) a ConvNet that accepts a local window of raw data as its input at each time step, and (4) a higher "feature fusion" (i.e., dense) layer to combine the LSTM's and ConvNet's outputs. On three univariate time series data sets, the TreNet outperforms the competing baselines including those based on its constituent parts (LSTM + trend inputs, CNN).

Strengths:
- A very interesting problem setting that can plausibly be argued to differ from other sequential modeling problems in deep learning (e.g., video classification). This is a nice example of fairly thoughtful task-driven machine learning.
- Accepting the author's assumptions as true for the moment, the proposed architecture seems intuitive and well-designed.

Weaknesses:
- Although this is an interesting problem setting (decisions driven by trends and changes), the authors did not make a strong argument for why they formulated the machine learning task as they did. Trend targets are not provided from "on high" (by data oracle) but extracted from raw data using a deterministic algorithm. Thus, one could just easily formulate this as plain time series forecasting problem in which we forecast the next 100 steps and then apply the trend extractor to convert those predictions into a trend. If the forecasts are accurate, so will be the extracted trends.
- The proposed architecture, while interesting, is not justified, in particular the choice to feed the extracted trends and raw data into separate LSTM and ConvNet layers that are only combined at the end by a shallow MLP. An equally straightforward but more intuitive choice would have been to feed the output of the ConvNet into the LSTM, perhaps augmented by the trend input. Without a solid rationale, this unconventional choice comes across as arbitrary.
- Following up on that point, the raw->ConvNet->LSTM and {raw->ConvNet,trends}->LSTM architectures are natural baselines for experiments.
- The paper presupposes, rather than argues, the value of the extracted trends and durations as inputs. It is not unreasonable to think that, with enough training data, a sufficiently powerful ConvNet->LSTM architecture should be able to learn to detect these trends in raw data, if they are predictive.
- Following up on that point, two other obvious baselines that were omitted: raw->LSTM and {raw->ConvNet,trends}->MLP. Basically, the authors propose a complex architecture without demonstrating the value of each part (trend extraction, LSTM, ConvNet, MLP). The baselines are unnecessarily weak.

One thing I am uncertain about in general: the validity of the practice of using the same LSTM and ConvNet architectures in both the baselines and the TreNet. This *sounds* like an apples-to-apples comparison, but in the world of hyperparameter tuning, it could in fact disadvantage either. It seems like a more thorough approach would be to optimize each architecture independently.

Regarding related work and baselines: I think it is fair to limit the scope of in-depth analysis and experiments to a set of reasonable, representative baselines, at least in a conference paper submitted to a deep learning conference. That said, the authors ignored a large body of work on financial time series modeling using probabilistic models and related techniques. This is another way to frame the above "separate trends from noise" problem: treat the observations as noisy. One semi-recent example: J. Hernandez-Lobato, J. Lloyds, and D. Hernandez-Lobato. Gaussian process conditional copulas with applications to financial time series. NIPS 2013.

I appreciate this research direction in general, but at the moment, I believe that the work described in this manuscript is not suitable for inclusion at ICLR. My policy for interactive review is to keep an open mind and willingness to change my score, but a large revision is unlikely. I would encourage the authors to instead use their time and energy -- and reviewer feedback -- in order to prepare for a future conference deadline (e.g., ICML).

[Official Review · AnonReviewer1 · rating 4 · confidence 4 · 16 Dec 2016]
**Promising architecture but insufficient experiments**

1) Summary

This paper proposes an end-to-end hybrid architecture to predict the local linear trends of time series. A temporal convnet on raw data extracts short-term features. In parallel, long term representations are learned via a LSTM on piecewise linear approximations of the time series. Both representations are combined using a MLP with one hidden layer (in two parts, one for each stream), and the entire architecture is trained end-to-end by minimizing (using Adam) the (l2-regularized) euclidean loss w.r.t. ground truth local trend durations and slopes.
 
2) Contributions

+ Interesting end-to-end architecture decoupling short-term and long-term representation learning in two separate streams in the first part of the architecture.
+ Comparison to deep and shallow baselines.

3) Suggestions for improvement

Add a LRCN baseline and discussion:
The benefits of decoupling short-term and long-term representation learning need to be assessed by comparing to the popular "long-term recurrent convolutional network" (LRCN) of Donahue et al (

[Author Response · Tao Lin · 14 Jan 2017]
**Revised Paper Uploaded**

Dear Reviewers,

We have uploaded a revised version of the paper, details are as follow:

* We have extended the experiment section by adding results from two new baselines suggested by the earlier review. One is the naive approach which takes the duration and slope of the last trend as the prediction for the next one. The other one is based on the cascade structure of ConvNet and LSTM in which feeds the features learned by ConvNet over time series to an LSTM and obtains the prediction from the LSTM.

* We have fixed some small typos.

We are also going to further reorganize the paper based on the comments.

[Final Decision · Program Chairs · 06 Feb 2017]
**ICLR committee final decision**

I appreciate the authors putting a lot of effort into the rebuttal. But it seems that all the reviewers agree that the local trend features segmentation and computation is adhoc, and the support for accepting the paper is lukewarm.
 
 As an additional data point, I would argue that the model is not end-to-end since it doesn't address the aspect of segmentation. Incorporating that into the model would have made it much more interesting and novel.